# Loss of the KN Motif and AnKyrin Repeat Domain 1 (KANK1) Leads to Lymphoid Compartment Dysregulation in Murine Model

**DOI:** 10.3390/genes14101947

**Published:** 2023-10-16

**Authors:** Marwa Almosailleakh, Sofia Bentivegna, Samuele Narcisi, Sébasitien J. Benquet, Linn Gillberg, Carmen P. Montaño-Almendras, Simonas Savickas, Erwin M. Schoof, Amelie Wegener, Hérve Luche, Henrik E. Jensen, Christophe Côme, Kirsten Grønbæk

**Affiliations:** 1Department of Hematology, Rigshospitalet, 2100 Copenhagen, Denmark; marwa.almosailleakh@bric.ku.dk (M.A.);; 2Biotech Research and Innovation Centre (BRIC), Faculty of Health and Medical Sciences, University of Copenhagen, 4072 Copenhagen, Denmark; 3Department of Biotechnology and Biomedicine, Technical University of Denmark (DTU), 2800 Kongens Lyngby, Denmark; 4Janvier Labs, 53010 CEDEX Laval, France; 5Centre d’Immunophénomique—CIPHE (PHENOMIN), Aix Marseille Université (UMS3367), Inserm (US012), CNRS (UAR3367), 13397 Marseille, France; 6Department of Veterinary and Animal Sciences, Faculty of Health and Medical Sciences, University of Copenhagen, 1958 Frederiksberg C, Denmark

**Keywords:** myelodysplastic neoplasms (MDS), germline (GL), KANK1, murine model, lymphoid dysregulation

## Abstract

The KN Motif and AnKyrin Repeat Domain 1 (*KANK1*) is proposed as a tumour suppressor gene, as its expression is reduced or absent in several types of tumour tissue, and over-expressing the protein inhibited the proliferation of tumour cells in solid cancer models. We report a novel germline loss of heterozygosity mutation encompassing the *KANK1* gene in a young patient diagnosed with myelodysplastic neoplasm (MDS) with no additional disease-related genomic aberrations. To study the potential role of KANK1 in haematopoiesis, we generated a new transgenic mouse model with a confirmed loss of KANK1 expression. KANK1 knockout mice did not develop any haematological abnormalities; however, the loss of its expression led to alteration in the colony forming and proliferative potential of bone marrow (BM) cells and a decrease in hematopoietic stem and progenitor cells (HSPCs) population frequency. A comprehensive marker expression analysis of lineage cell populations indicated a role for *Kank1* in lymphoid cell development, and total protein analysis suggests the involvement of *Kank1* in BM cells’ cytoskeleton formation and mobility.

## 1. Introduction

Myelodysplastic neoplasms (formerly known as myelodysplastic syndrome or MDS) refers to a group of clonal haematopoietic disorders characterised by morphological dysplasia, ineffective haematopoiesis, peripheral-blood cytopenia, and a high risk of progression to acute myeloid leukemia (AML) [1,2]. The disease generally has a sporadic onset predominantly affecting the elderly, while remaining rare in children and young adults. The exact pathogenesis of MDS is very heterogenous and varies among patients with no clear etiological and causative factors. The disease is believed to develop by a complex combination of somatic and germline genetic abnormalities of haematopoietic stem and progenitor cells (HSPCs), epigenetic alterations, exogenous bone marrow (BM) microenvironment changes, and immune dysregulation [3,4,5,6,7].

We identified a novel germline deletion resulting in a loss of heterozygosity (LOH) mutation encompassing the KN Motif and Ankyrin Repeat Domain 1 (*KANK1*) gene in a young patient diagnosed with MDS and the patient’s healthy father. The patient had no additional MDS-related mutations or chromosomal aberrations, but suffered from severe anaemia, which despite numerous treatment attempts, was only cured by allogeneic stem cell transplantation. The KANK1 protein has a unique structure of coiled-coil and KN motifs in the N-terminus and 5 ankyrin repeat domains at the C-terminus and is reported to play key roles in the control of cytoskeleton formation, cell polarity, and migration by regulating actin filament polymerisation [8,9,10]. Linkage analysis of a multigenerational familial form of congenital cerebral palsy characterised by quadriplegia and mental retardation attributed the disease to inherited germline microdeletions in *KANK1* [11]. In solid cancer, the expression of KANK1 was reported to be either completely missing or downregulated in tumour tissues, and over-expressing the protein inhibited the proliferation of malignant cells, suggesting a tumour suppressor function [12,13,14,15,16]. A t(5;9) translocation resulting in a chimeric fusion protein of the platelet-derived growth factor receptor β gene (*PDGFRB*) and *KANK1* was detected in a patient with a myeloproliferative neoplasm (MPN) characterised with severe thrombocythemia [17]. Furthermore, defects in cytoskeletal proteins are reported to contribute to immunological dysfunction due to their essential role in almost all stages of immune system function, from haematopoiesis and immune cell development, through to recruitment and migration [6,18]. In this paper, we study the role of KANK1 in normal haematopoiesis and MDS. We report that upon the loss of KANK1 expression, we observe an alteration in the colony forming and proliferative capacity of murine BM cells and a decrease in HSPCs population frequency. Comprehensive marker expression analysis revealed a dysregulation of immune cell development.

## 2. Materials and Methods

### 2.1. The Young MDS Patient

A novel germline deletion of ~190,000 bp resulting in a LOH mutation encompassing the *KANK1* gene on chromosome 9 (9p24.3, 535,417–725,581) was identified in a 28-year-old patient with MDS (refractory cytopenia with multilineage dysplasia) at Rigshospitalet, Denmark, using SNP array analysis (Appendix A). No additional MDS-related mutations or structural alterations were detected in the patient’s sample. The patient suffered from severe anaemia and clinical investigation suggested he developed MDS following severe infection. The same LOH mutation was identified in the patient’s healthy father, but not the mother.

### 2.2. Generation of Mice

*Kank1^−/−^* mice were generated in the core facility for transgenic mice (University of Copenhagen, Denmark) as described previously [19]. Briefly, a mixture of in vitro transcribed sgRNA and Cas9 mRNA was microinjected into the cytoplasm and nucleus of one cell-stage embryo. Zygotes were then cultured in KSOM and transferred the same day to recipient pseudopregnant CD1 females, which carried them to term. Pups were genotyped by PCR, followed by verification with Sanger sequencing. F0 pups with confirmed editing were backcrossed into wild-type (C57BL/6NRj) to establish the transgenic mouse colony.

All mouse studies were conducted in adherence to protocols approved by the Danish Animal Ethical Committee. Mice were bred and housed locally at the Biotechnology and Innovation Center (BRIC) at the University of Copenhagen.

### 2.3. Genotyping

Genomic DNA was purified using the NaOH extraction protocol [20]. Briefly, 500 µL of 50 mM NaOH is added to ear biopsies in 1.5 mL Eppendorf tubes and incubated at 95 °C for 20 min with shaking. A volume of 50 µL of 1 M Tris-HCL buffer is then added to the tube and mixed by vortexing. A volume of 3 µL of undiluted DNA mixture is used for PCR using EconoTaq PLUS GREEN 2× master mix (Lucigen, cat. no. 30033-1); Fp: 5′GAAAGACCCGTATTTTGTGG3′ and Rp: 5′GGATTGGGGTTGAAGTAACG3′; PCR program: initial denaturation, 94 °C, 5 min; denaturation, 95 °C, 30 s; annealing, 58 °C, 30 s; extension, 72 °C, 1 min; extension, 72 °C, 5 min; for 40 cycles. The bands were resolved in 4% agarose gel and visualised using the Gel Doc XR+ system (Bio-rad, Hercules, CA, USA).

### 2.4. Analysis of the Mouse Phenotype

Mice were sacrificed using neck dislocation, and organs were removed and fixed in 4% paraformaldehyde (PFA) solution overnight at 4 °C and then submerged in 70% ethanol. PFA-embedded tissue sections were stained with hematoxylin and eosin (H&E). Blood was collected by cardiac puncture and counts were determined using an Element HT5 instrument (HESKA).

### 2.5. Colony Forming Assay

For total BM analysis, approximately 4 × 10^4^ cells were plated in M3434 methylcellulose media (Cat. no. 03434 Methocult, StemCell Technologies, Vancouver, BC, Canada), and incubated at 37 °C and scored after 5–7 days. Colony images were taken using an Evos FL Auto2 microscope (Invitrogen, ThermoFisher Scientific, Waltham, MA, USA) with ×2, ×4, and ×10 magnification. Cells were washed in warm PBS, resuspended, and counted with cell Nucleocounter NC-250^TM^ (ChemoMetec, Copenhagen, Denmark) and replated into fresh methylcellulose; 0.25–0.5 × 10^6^ cells were used for Cytospins preparations by 3 min centrifugation at 300 rpm (Shandon Cytospin 3) in a TPX reusable funnel (Lab Teamet, Helsingborg, Sweden) and non-coated cytoslides (Cat. no. 5991051, Epredia, NH, USA). Cytopins were stained with May-Grünwalds Wright Giemsa solution (Cat. no. HX41658024, Merck, Darmstadt, Germany).

### 2.6. RT-PCR

Total RNA was extracted using the Allprep DNA/RNA/miRNA universal kit (Cat. no. 80224, Qiagen, Hilden, Germany) according to the manufacturer’s protocol. First strand cDNA was synthesized using the TaqMan Reverse Transcription kit (N8080234, Applied biosystems, Foster City, CA, USA). Quantitative PCR was performed using PowerTrackTM SYBR Green Master Mix for qPCR (Cat. no. A46109, ThermoFisher Scientific, Waltham, MA, USA) and an ABI prism 7500 sequence instrument. Ct values were normalized to *Gapdh* expression and relative expression was quantified using the 1/dCt method. Primer sequences: *Gapdh* Fp: 5′CATCTTCTTGTGCAGTGCCAG3′, Rp: 5′GGCAACAATCTCCACTTTGCC3′, *Kank1* Fp: 5′ CCAGTGAACACGGGCATGTG3′, Rp: 5′ TTCGAGAAGTTGAGGTGGGC3′, *Kank1-del* Fp: 5′TCAGCTGTGGAGACGTGAAC3′.

### 2.7. Flow Cytometry Analysis

Cells in suspension were blocked with Mouse BD Fc Block™ (BD Biosciences, San Jose, CA, USA) and incubated (30 min at 4 °C in the dark) in PBS 2% FBS containing Brilliant Stain Buffer Plus (BD Biosciences, San Jose, CA, USA) with the following antibodies: CD127-BUV737 (SB/199), CD34-BV786 (RAM34), CD16/32-BB700 (Ab93), CD3-BV605 (145-2C11), CD8a-BV711 (53–6.7), CD4-BV786 (GK1.5), NK-1.1-FITC (PK136), CD19-BB700 (1D3), CD11b-PE (M1/70) and Ly-6G-PE-Cy7 (1A8) from BD Biosciences; CD135-BV421 (A2F10), CD150-BV605 (TC15-12F12.2), CD48-PE-Cy7 (HM48-1) and lineage cocktail-FITC from BioLegend (cat. no. 78022); CD117-PE (ACK2) and Sca-1-APC (D7) from Invitrogen. Stained cells were analysed on LSR Fortessa II (BD Biosciences, San Jose, CA, USA). Data were analysed with FlowJo software (version 10.0.8 or 10.2). Detailed information about the FACS gating strategies can be found in the Appendix A.

Multiparametric immunophenotyping was performed at the CIPHE-PHENOMIN (Inserm, US012, Marseille, France) flow cytometry facility. Peripheral Blood (PB) was collected by submandibular puncture using Microvette^®^ 500 K3 EDTA (Sarstedt). PB Leukocytes were analysed using a Lyse No Wash protocol using 1× FACS Lysing Solution (BD Biosciences, San Jose, CA, USA). Leukocytes from spleen and thymus were extracted as described on the IMPRESS protocol (https://www.mousephenotype.org/impress/protocol/174/7, accessed on 1 August 2021). Briefly, organs were disrupted with 600 Mendel Unit Collagenase D (Roche Life Science, Basel, Switzerland) and 30 μg DNAse I (Sigma, Marlborough, MA, USA) for 20 min at RT. The cell suspension was filtered and counted. Red blood cells were lysed by the addition of ACK lysis solution (eBioscience, San Diego, CA, USA) for 1 min at RT. Cells were then incubated with the anti-CD16/32 (2.4G2) antibody to block Fc receptors for 10 min at 4 °C. DAPI (Invitrogen, Waltham, MA, USA) was used for dead cell exclusion. The multiparameter FACS acquisition of stained cells was performed on a Fortessa LSRII SORP or Canto 10C system (BD Biosciences, San Jose, CA, USA). Data analysis was performed using FACSDiva 9.01 (BD Biosciences, San Jose, CA, USA) software. The gating strategies and antibodies used for immunophenotyping can be found in Appendix A, respectively.

### 2.8. Total Protein Mass Spectrometry Analysis

Lysate preparation and digestion were performed as described in [21]. Briefly, a total of 0.5 × 10^6^ *Kank1^+/+^* and *Kank1^−/−^* BM cells were lysed using 20 µL of lysis buffer (6 M Guanidinium Hydrochloride, 10 mM TCEP, 40 mM CAA, 50 mM HEPES pH8.5) and boiled at 95 °C for 5 min, after which they were sonicated on high for 3 × 10 s in a Bioruptor sonication water bath (Diagenode, Liege, Belgium) at 4 °C. Protein samples were digested in buffer and trypsin (10% Acetonitrile, 50 mM HEPES pH 8.5, trypsin, MS grade, Promega, Madison, WI, USA) in a 1:100 (enzyme to protein) ratio and incubated overnight at 37 °C. Enzyme activity was quenched by adding 2% trifluoroacetic acid (TFA) to a final concentration of 1%. Prior to mass spectrometry (MS) analysis, the peptides were desalted [22] and later eluted into Eppendorf tubes using 40% Acetonitrile, 0.1% formic acid. The eluted peptides were concentrated and then re-constituted in 1% TFA, 2% Acetonitrile for MS analysis.

### 2.9. MS Data Acquisition

For each sample, peptides were eluted over a 140-min gradient at 250 nL/min using the Exploris480 instrument (Thermo Fisher Scientific, Waltham, MA, USA) run in a DD-MS2 setting with multi-CV FAIMS. For each CV, full MS spectra were collected at a resolution of 60,000, whereas MS2 spectra were obtained at a resolution of 15,000. Dynamic exclusion was set to 60 s, and ions with a charge state < 2 or unknown were excluded. The MS performance was verified for consistency by running complex cell lysate quality control standards, and chromatography was monitored to check for reproducibility. The MS data have been deposited to the ProteomeXchange Consortium (http://proteomecentral.proteomexchange.org, accessed on 1 September 2023) via the PRIDE partner repository with the dataset identifier PXD PXD045032.

### 2.10. Label-Free Quantitative Proteomics Analysis

The raw files were analysed using Proteome Discoverer 2.4. Label-free quantitation (LFQ) was enabled in the processing and consensus steps, and spectra were matched against the Mus Musculus database obtained from Uniprot. All results were filtered to a 1% FDR, and protein quantitation was performed using the built-in Minora Feature Detector and Precursor Ions Quantifier.

### 2.11. Targeted KANK1 Protein Quantification

Sample preparation and analysis were performed as previously described [23,24]. Briefly, one hundred micrograms of protein were cleaved into short peptides using trypsin in a final ratio of 1:50 (protease: protein *w*/*w*) and injected on the EvoSep one MS instrument. The MS instrument was operated in parallel reaction monitoring (PRM) mode, scanning for the following peptide sequences: ICLNTLQHDWFR, LLLDADVCNVDHQNK, DIAVLLYAHLNFSK, AQSPSTPR, TSPGPTHR. MS1 scans were collected at a 60,000 resolution, scan range of 375–1500, and a maximum injection time (IT) of 50 ms, 4 × 10^5^ AGC target value. MS2 precursors were isolated with 0.8 Da isolation windows, collected for 86 ms IT, AGC target of 5 × 10^5^, fragmented with 5% stepped collision energy (HCD 28), and the resulting fragment ions were scanned at 50,000 resolution in the Orbitrap, with a total cycle time of 3 s. The resulting data were inspected, quantified, and visualised in Skyline [25].

## 3. Results

### 3.1. Loss of KANK1 Reduced the Frequency of Hematopoietic Stem and Progenitor Cells (HSPCs) in BM of Adult Mice

To remodel the LOH identified in the young patient (Appendix A) and study the role of *Kank1* in haematopoiesis, we created a knockout transgenic mouse line (*Kank1^−/−^*) using CrispR-Cas9 editing technology (Appendix A). First, we validated the loss of *Kank1* exon5 mRNA and total protein expression by q-PCR and targeted MS analysis (Appendix A). To characterise the mouse line and check for haematological malignancies, littermate mice of all possible genotypes (*Kank1^+/+^*, *Kank1^+/−^* and *Kank1^−/−^*) were followed for an observational period of two years, where the surviving mice were sacrificed and autopsied, and their tissue was analysed (Figure 1a,b). We noted no difference in the survival rate of the *Kank1^−/−^* mice compared to wild-type (*Kank1^+/+^*) or heterozygous (*Kank1^+/−^*) (Figure 1a), nor the development of any signs of disease (Figure 1b). Furthermore, the peripheral blood parameters of all sacrificed mice, including white blood count (WBC), red blood count (RBC), and platelets (PLT), remained within the normal range (Appendix A). The examination of the BM cells from adult *Kank1^−/−^* mice (19 weeks) for the expression of various stem and progenitor cell markers showed a significantly reduced number of cells in the lineage marker negative, ckit^+^/Sca1^+^ (LSK), and multi-potent progenitor1 (MPP1) compartments (Figure 1c(i,ii)). However, the number of all the other progenitors, including the common lymphoid and myeloid progenitors (CLP and CMP, respectively), remained similar (Figure 1c(iii,iv), Appendix A). We expected this dysregulation in *Kank1^−/−^* stem and/or progenitor BM compartments to become more pronounced upon ageing; however, surprisingly, we did not detect any significant difference in population frequency in older mice (53–58 weeks, Figure 1d, Appendix A). Furthermore, the frequency of all major lymphoid cell types, including the B-, T-, and NK- cells and macrophages in the BM of *Kank1^+/+^* and *Kank1^−/−^*, were similar in adults and old mice (Supplementary Appendix A).

To further characterise the effect of KANK1 loss on BM stem cell self-renewal, proliferation and differentiation capacities, we serially plated total BM cells from wild-type (*Kank1^+/^*^+^) and knockout (*Kank1^−/−^*) mice in methylcellulose media (MC) promoting the growth of cells of the myeloid lineage (Figure 1e,f, Appendix A). In agreement with our previous results regarding the reduction in the stem and progenitor cell population in *Kank1^−/−^* mice, the BM cells of knockout mice formed reduced number of colonies in MC with smaller and more compact morphology, indicating a reduction in stem cell number and proliferative capacity (Figure 1e,f). However, we detected no differences in the myeloid (Mac1^+^ Gr1^+^) or the erythroid (cKit^+^ CD71^+^) lineage markers expression, suggesting that the loss of KANK1 did not alter the cells’ differentiation potential (Appendix A).

### 3.2. Loss of KANK1 Leads to Lymphoid Compartment Dysregulation

To further understand the role of KANK1 in steady state haematopoiesis, we performed a comprehensive flow cytometry analysis of the myeloid and lymphoid compartments in old (68–71 weeks) and young (6–7 weeks old) wild-type and knockout mice (Figure 2a–c, Appendix A). Principle component analysis (PCA) of all mice based on markers expression showed a dominance of the age variable over mutational status (Figure 2a). In the peripheral blood of old knockout mice, we detected a significant decrease in the monocytic and B1B lymphocytic cell population when compared to wild-type mice (Figure 2b(i)). Flow cytometry analysis also showed reduced Treg and macrophage populations in knockout splenocytes from old mice (Figure 2b(ii)). In the peripheral blood of young mice, the activated CD4 CD44hi fraction of T-cells was significantly reduced in knockout mice, whereas in the spleen, similar to the old mice, we detected a modest reduction in the Treg subpopulation (Figure 2c(i,ii)). In the thymus of young mice, double positive CD3^+^/CD69^+^ and TN3b and TN1ab T-cells were all significantly decreased in knockout mice compared to wild-type, coupled with an increase in the TN4 and double negative (DN) populations (Figure 2c(iii)). Next, using MS, we performed a global protein content analysis of BM cells from wild-type and knockout mice (Figure 2d,e). In total, we found significant changes in 14 proteins (FC > 0.5, adj. *p* < 0.05), 12 of which were reduced in knockout mice compared to wild-type and two with an increased expression (Figure 2d). In accordance with the proposed role of KANK1 in actin polymerisation, most of the reduced proteins were associated with cytoskeleton formation such as myosin-1 (MYH1), tropomysin-1 (TDM1), and annexin (ANXA2), and extracellular matrix components such as collagenα (CO1A1) and matrix metalloproteinase-9 (MMP-9) (Figure 2d). The pathway analysis of significantly altered proteins revealed protein pathways associated with muscle contraction and solid tumours (Figure 2e). Taken together, comprehensive flow cytometry analysis indicates that the loss of KANK1 seems to have more of an effect on cellular subtypes of the lymphoid compartment rather than the myeloid in both young and ageing mice, whereas total protein analysis confirms a role for KANK1 in cell mobility and cytoskeleton formation.

## 4. Discussion

Despite its reported role as a tumour suppressor gene in multiple solid cancer models, the role of KANK1 in blood cells and haematological malignancy is limited and not well characterised. To remodel the LOH of *KANK1* identified in a young MDS patient and decipher the role of KANK1 in steady-state haematopoiesis and MDS, we generated a *Kank1* knockout mouse line using CrispR-Cas9 editing. Although we confirmed the loss of KANK1 expression on both the mRNA and protein levels in two different tissues, the liver and BM, respectively, the *Kank1^−/−^* mice were phenotypically indistinguishable from their wild-type littermate controls and did not develop any signs of distress or disease (Figure 1). Interestingly, in a recent publication, an attempt to generate a *Kank1* knockout mouse line by deleting the entire exon5 produced a mosaic founder mouse and, thus, failed to breed the deletion through the germline [26]. The authors’ second approach of inserting a strong stop codon downstream of the initiating AUG codon also failed to result in the loss of KANK1 protein in the edited mice. The authors speculate that this could be due to the presence of a second in-frame AUG start codon downstream of the editing site initiating the translation of a shorter form of protein. To support their hypothesis, they discuss that whereas the human *KANK1* gene possesses more than 20 different variants, only two variants are reported for mice [26]. Notably, we found it to be quite challenging to design a unique protein targeting sequence to perform the targeted MS analysis of KANK1 in mouse BM cells, which we thought was due to the structural similarities of all KANK proteins specially at the C-terminal, but it could also be due to the generation of a truncated form of the KANK1 with complete or partial activity [16].

The loss of KANK1 resulted in a reduced number of stem and progenitor cells in adult knockout mice compared to controls, which was surprisingly resolved upon ageing. BM HSPCs from knockout mice also had lower colony formation and proliferation potential in MC compared to controls but retained a comparable differentiation potential (Figure 1). The extensive marker expression analysis of knockout mice and controls revealed a mild dysregulation in cells of the lymphocytic lineage, presented as a reduction in the Treg population in the spleen of both young and old mice and an increase in the premature T-cell population in the thymus of young mice (Figure 2). To obtain an overview on global protein expression changes in BM cells upon the loss of KANK1, we performed MS analysis and identified a reduced expression of several proteins associated with cytoskeleton formation, which was in accordance with the KANK1 reported role in regulating cytoskeleton organisation, cell polarity, and migration (Figure 2) [9,16,27,28,29,30]. Interestingly, after the initial step of T cell-receptor (TCR) triggering by peptide complexes on antigen presenting cells (APC), the T-cell undergoes extensive cytoskeletal changes and the modulation of actin and microtubule remodelling. Defects in actin effectors that couple TCR activation to actin rearrangements result in T-cell activation impairment and, thus, a defective immunological response [18,31,32,33]. As the association between a dysregulation of the immune system and development of MDS is well established [6,7,34], we speculate that defective TCR activation due to the loss of KANK1 and dysregulation in actin remodeling might have led to a defective immunological response in the patient during an infection incident, triggering the onset of MDS. To examine this hypothesis, we need to perform more detailed functional analysis to assess the activity of *Kank1^−/−^* immune cells in response to infections in in vitro and in vivo experimental settings, which would be interesting to address in future studies.

## Figures and Tables

**Figure 1 genes-14-01947-f001:**
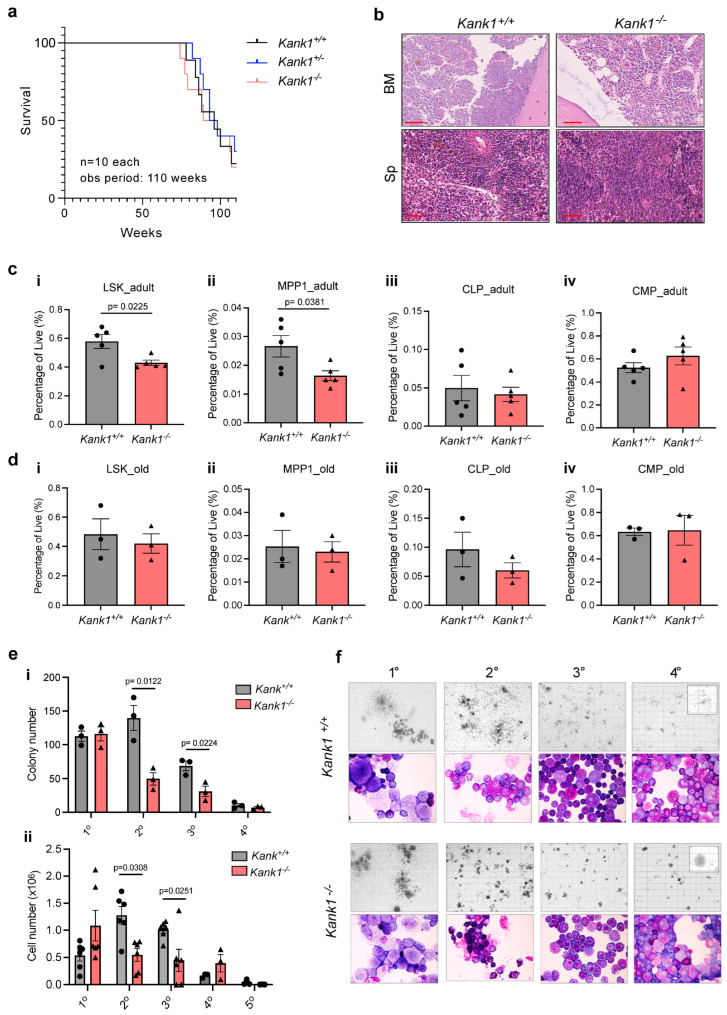
KANK1 knockout in mice did not lead to development of a myeloproliferative disease. (**a**) A Kaplan–Meier plot comparing the survival of *Kank1^+/+^, Kank1 ^+/−^*, and *Kank1^−/−^* mice. Mice were followed for 110 weeks (*n* = ten per genotype). (**b**) HE-staining of bone marrow (BM) and spleen (Sp) histology images from old (80–120 weeks) *Kank1^+/+^* and *Kank1^−/−^* mice (*n* = six per genotype). Scale bar (red) represents 50 μM. (**c**) Bar plots showing the quantification (%) of stem (LSK and MPP1, (**i**,**ii**)) and progenitor (CLP and CMP, (**iii**,**iv**)) cell populations in the BM from adult (19 weeks) *Kank1^+/+^* and *Kank1^−/−^* mice *(n* = five per genotype, *n* = four females, and *n* = one male). (**d**) Bar plots showing the quantification (%) of stem (LSK and MPP1, (**i**,**ii**)) and progenitor (CLP and CMP, (**iii**,**iv**)) cell populations in the BM from old (53–58 weeks) *Kank1^+/+^* and *Kank1^−/−^* mice (*n* = three per genotype, *n* = two females, and *n* = one male per genotype). (**e**) Bar plot showing the number of colonies (**i**) and cell progeny (**ii**) by 40,000 total BM cells from *Kank1 ^+/+^* and *Kank1^−/−^* from methylcellulose plates (M3434) at each replating (5–7 days, *n* = three for colony number, *n* = six for cell number per genotype). (**f**) Representative microscopic images of BM cell colonies (2.5×) in methylcellulose (M3434) and Wright-Giemsa-stained (100×) cytospin preparations of *Kank1 ^+/+^* (top panel) and *Kank1^−/−^* (lower panel). These data represent one out of six independent experiments. Values are presented as individual points, bar graphs represent the mean value of biological replicates, error bars as standard error of the mean. Statistical significances tested with unpaired two-tailed *t*-test.

**Figure 2 genes-14-01947-f002:**
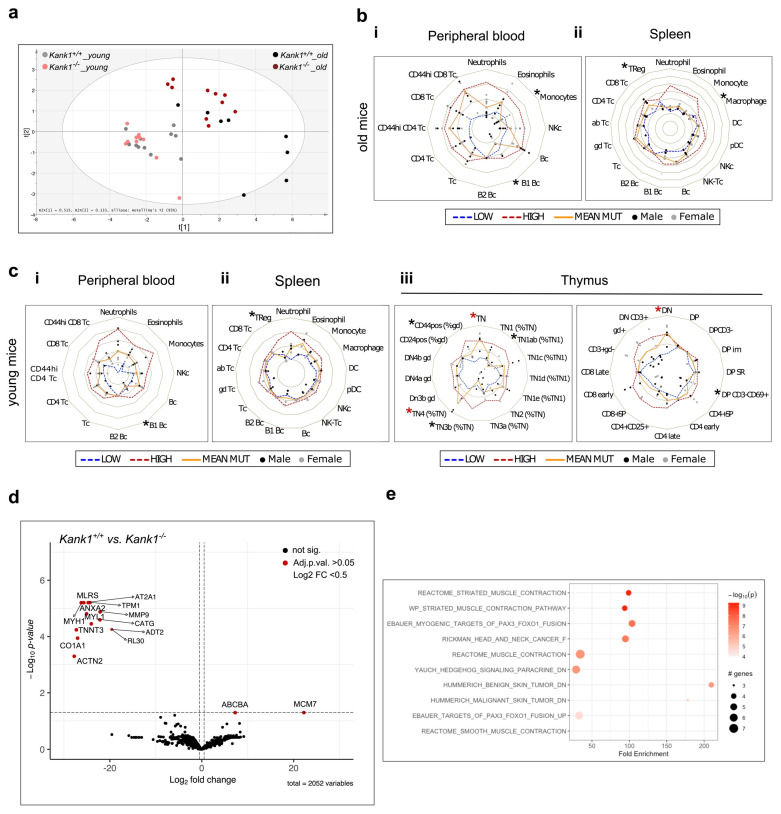
Loss of KANK1 leads to HSPCs and lymphoid compartment dysregulation in mice. (**a**) Principal component analysis (PCA) of mice based on genotype (*Kank1^+/+^* vs. *Kank1^−/−^)* and age (young, 7 weeks vs. old, 70 weeks). (**b**) Flow cytometry radar plots of comprehensive immuno-phenotyping analysis of haematopoietic cells in peripheral blood (**i**) and spleen (**ii**) from old (70 weeks) *Kank1^+/+^* and *Kank1^−/−^* mice (*n* = eight per genotype). (*) Marks reduced populations, whereas (*) marks enhanced populations. (**c**) Flow cytometry radar of comprehensive immuno-phenotyping analysis of haematopoietic cells in peripheral blood (**i**), spleen (**ii**), and thymus (**iii**) from young (7 weeks) *Kank1^+/+^* and *Kank1^−/−^* mice (*n* = eight per genotype). (*) Marks reduced populations, whereas (*) marks enhanced populations. (**d**) Volcano plot of differential protein expression of *Kank1^+/+^* vs. *Kank1^−/−^* BM cells (*n* = four per genotype, FDR < 0.05, adjusted *p*-value < 0.05). Labels are shown for proteins with logFC > 0.5. (**e**) Pathway analysis of significantly altered proteins in bone marrow (BM) of *Kank1^+/+^* vs. *Kank1^−/−^* mice. In flow cytometry radar plots, each circle (black: male; grey: female) represents a value obtained in independent *Kank1^+/+^* (WT) and *Kank1^−/−^* (KO) mice (value measured in (KO)/(Mean WT) − 1), orange line: average variation of subset proportion in KO compared with WT animals ((Mean KO)/(Mean WT) − 1); blue dashed line: lower limit of WT value dispersion ((Mean WT − 1 SD)/(Mean WT) − 1); red dashed line: higher limit of WT value dispersion ((Mean WT + 1 SD)/(Mean WT) − 1). Populations were defined according to the expression of markers detailed in Appendix A. Values expressed in Asinh ratio. Values are presented as individual points.

## Data Availability

Patient’s data cannot be deposited in public repositories, as it considered sensitive personal data according to Danish law and the European Union General Data Protection Regulation (GDPR) and cannot be shared with third parties without prior approval. To access the data, please contact the corresponding author at kirsten.groenbaek@regionh.dk. Access can only be granted for research purposes and only if a data processor or data transfer agreement can be made in accordance with Danish and European law. The expected timeframe from response until access is granted is ~6 months. The total protein analysis data is deposited to the ProteomeXchange Consortium (http://proteomecentral.proteomexchange.org, accessed on 1 September 2023) via the PRIDE partner repository with the dataset identifier PXD PXD045032.

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
