# Peer review of "Loss of the KN Motif and AnKyrin Repeat Domain 1 (KANK1) Leads to Lymphoid Compartment Dysregulation in Murine Model"

_genes, 2023, doi:10.3390/genes14101947_

Round 1
Reviewer 1 Report
The manuscript presented the phenotypic analysis of hematopoietic tissues in KANK1 KO mice. The authors reported a KANK1 mutation in a MDS patient and wanted to study the function of KANK1 in normal hematopoiesis and potential in MDS pathogenesis. The experimental design is good and strong. However, due to the limited number of mice were analyzed in each group, the conclusion is not solid. Gender was not taken into the consideration of data analysis. Although, the role of infection and inflammation in the pathogenesis of MDS was discussed in the introduction section, however the mice were only analyzed under homeostatic condition. Thus, more data is required to fully support the conclusion.
Major concerns:
1. In figure 1C and 1D, LSK and MPP1 were compared between WT and KO mice, only 3 mice were analyzed in each genotype. Although difference in young adult was observed, it is difficult to reach a conclusion of HSPC reduction due to the small number of mice studied. It is unclear the gender of the mice in each group. It was known that male mice normally have higher LSK cells then that in female mice.
2. Figure 2.B and C. Flow cytometry radar plots: due to the busy lines, without detailed information, it is difficult to determine which lines represent which.
3. Naïve T-cells (TN3b &TN1ab) were all significantly decreased in knockout mice compared to wild-type, coupled with an increase in the TN4 and double negative (DN) populations. Naïve T-cells normally refer to mature T cells in peripheral lymph tissues. It is difficult to understand what Naïve T-cells (TN3b &TN1ab) refer to.
4. In discussion: Despite its reported role as a tumor suppressor gene in multiple solid cancers models, the specific role of the KANK1 gene in leukemia is limited, and not well characterized. This manuscript did not study the role of KANK1 in leukemia.
Author Response
Reviewer 1:
The manuscript presented the phenotypic analysis of hematopoietic tissues in KANK1 KO mice. The authors reported a KANK1 mutation in a MDS patient and wanted to study the function of KANK1 in normal hematopoiesis and potential in MDS pathogenesis. The experimental design is good and strong.
We would like to thank the reviewer for his thorough reading of our Manuscript and for helping us to improve it.
Comment 1:
However, due to the limited number of mice were analyzed in each group, the conclusion is not solid. Gender was not taken into the consideration of data analysis.
We agree with the reviewer regarding the limited number of mice, and we have added two more mice per genotype to the analysis (n=5) and we also specified the gender of animals in the figure legend.
Although, the role of infection and inflammation in the pathogenesis of MDS was discussed in the introduction section, however the mice were only analyzed under homeostatic condition. Thus, more data is required to fully support the conclusion.
We agree with the reviewer that the role of infection was not explored in this manuscript but our suggestion that inflammation is a likely cause of disease development is based on the patient’s clinical data and the dysregulation we detect on the mouse model. We acknowledge that studying the effect of inflammation/infection in vivo using our mouse model would be extremely pertinent, however we consider this outside the scope of the current manuscript. Nevertheless, we have removed the sentences regarding the patient’s infection diagnosis from the introduction and added it to ‘’the young patient’’ section in the materials and methods.
Major concerns:
- In figure 1C and 1D, LSK and MPP1 were compared between WT and KO mice, only 3 mice were analyzed in each genotype. Although difference in young adult was observed, it is difficult to reach a conclusion of HSPC reduction due to the small number of mice studied. It is unclear the gender of the mice in each group. It was known that male mice normally have higher LSK cells then that in female mice.
In response to this comment, we have further analyzed two more adult mice per genotype and noted the gender of the mice in the figure legend which was equal between the two compared groups.
- Figure 2.B and C. Flow cytometry radar plots: due to the busy lines, without detailed information, it is difficult to determine which lines represent which.
We agree with the reviewer and apologize for the quality of the radar plots, and we have simplified and changed the colors in the plots and added a plot key to the figure. We have also written a detailed description of the plots in the figure legend.
- Naïve T-cells (TN3b &TN1ab) were all significantly decreased in knockout mice compared to wild-type, coupled with an increase in the TN4 and double negative (DN) populations. Naïve T-cells normally refer to mature T cells in peripheral lymph tissues. It is difficult to understand what Naïve T-cells (TN3b &TN1ab) refer to.
We thank the reviewer for his comment, and we provide the following explanation. TN3b and TN1ab are not naive cells but these are immature thymocytes. TN1ab = ETP (Early thymic Progenitor) following the Petrie scheme while TN3b = subset of immature thymocyte that succeded TCRb selection at the pre-TCR check point. We have removed the word ‘’naïve’’ from the manuscript.
- In discussion: Despite its reported role as a tumor suppressor gene in multiple solid cancers models, the specific role of the KANK1 gene in leukemia is limited, and not well characterized. This manuscript did not study the role of KANK1 in leukemia.
We agree with reviewer that we did not study the role of leukemia in our study and apologize for the misunderstanding. We have since changed the sentence to ‘’ Despite its reported role as a tumor suppressor gene in multiple solid cancers models, the role of KANK1gene in blood cells and hematological malignancy is limited, and not well characterized.’’
Reviewer 2 Report
Almosailleakh et al. try to decipher the function and role of Kank1 by analyzing a KANK1 knock-out mouse after they have identified a KANK1 mutation in a patient with a myelodysplastic syndrome (MDS).
Even though Kank1 is classified as a tumor suppressor gene/protein it seems to have different functions in different tissues. In this manuscript the authors have identified a loss of heterozygosity (LOH) mutation in a young patient with myelodysplastic syndrome.
Very little information is given about the identified KANK1 mutation making the interpretation on whether this is a gain-of function (GOF) mutation (arguing for an oncogene) or a loss-of function mutation (LOF) (arguing for a tumor suppressor gene) impossible which I important for the approach to create and analyse a knock-out mouse. No analysis to predict the effect of the identified mutation is provided using bioinformatic programs such as PolyPhen2 or programs alike.
No mRNA or protein data from the patient is available regarding the presence of KANK1 mRNA (nonsense-mediated decay?) or protein.
The mechanisms of the LOH is also not discussed in detail making it difficult to evaluate whether taking a knock-out approach is feasible to mimic what is supposedly causing the MDS in the patient.
This might sound trivial but in light of the fact that a translocation engaging the PDGFR and the KANK1 gene has previously been identified in hematologic malignancies and that these translocations normally result in fusion proteins and overexpression it seems to be important to add more information regarding the identified KANK1 mutation.
The pictures/text in Figure 1 and 2 are very blurred and almost impossible to read (especially Figure 2), so I would like to ask the authors to upload a new version.
Author Response
Reviewer 2:
Almosailleakh et al. try to decipher the function and role of Kank1 by analyzing a KANK1 knock-out mouse after they have identified a KANK1 mutation in a patient with a myelodysplastic syndrome (MDS). Even though Kank1 is classified as a tumor suppressor gene/protein it seems to have different functions in different tissues. In this manuscript the authors have identified a loss of heterozygosity (LOH) mutation in a young patient with myelodysplastic syndrome.
Very little information is given about the identified KANK1 mutation making the interpretation on whether this is a gain-of function (GOF) mutation (arguing for an oncogene) or a loss-of function mutation (LOF) (arguing for a tumor suppressor gene) impossible which I important for the approach to create and analyse a knock-out mouse. No analysis to predict the effect of the identified mutation is provided using bioinformatic programs such as PolyPhen2 or programs alike. No mRNA or protein data from the patient is available regarding the presence of KANK1 mRNA (nonsense-mediated decay?) or protein.
The mechanisms of the LOH is also not discussed in detail making it difficult to evaluate whether taking a knock-out approach is feasible to mimic what is supposedly causing the MDS in the patient
We would like to first thank the reviewer for taking the time to read and review our paper. We agree with the reviewer regarding the limited data we provided on the nature of detected KANK1 mutation, and we therefore added a new section in the materials and methods titled ‘’the young MDS patient’’ where we provide the exact genomic location and size of the deletion. We have also added 3 new Cytoscan HD. array plots to Supplementary Figure 1 (Supp.Fig 1a) indicating the deleted region identified in the patient and the healthy father, and the unaffected mother. Unfortunately, we could not perform any mRNA or protein expression experiments due to the limited patient’s sample supply, however, we believe that with such a large deletion a conclusion of loss of heterozygosity is justified.
This might sound trivial but in light of the fact that a translocation engaging the PDGFR and the KANK1 gene has previously been identified in hematologic malignancies and that these translocations normally result in fusion proteins and overexpression it seems to be important to add more information regarding the identified KANK1 mutation.
We thank the reviewer for his interest in understanding the KANK1 disease mechanism and we would like to provide the following clarification. In their 2011 paper, Medves et al, proposed the constitutive activation of STAT5 transcription factors independently of JAK kinases as the main underlying mechanism of disease. We therefore suggest that although resulting in similar disease development, the role of KANK1 in normal lymphopoiesis and as tumour suppressor protein is probably via its function in cytoskeleton regulation and control and not due to abnormal STAT5 signaling. We have also added more information regarding the identified mutation in the materials and methods and in (Supp.Fig1a).
The pictures/text in Figure 1 and 2 are very blurred and almost impossible to read (especially Figure 2), so I would like to ask the authors to upload a new version.
We really apologize to the reviewer for this un-intentional mistake and for poor quality of the figures due to compression while saving, and we have amended this issue and provided a revised manuscript with sharp and clear figures.
Round 2
Reviewer 1 Report
The authors addressed all my concerns. However, I think it is better if the authors can compare the response of KO and WT BM cells to inflammatory cytokine treatment in CFU assays.
Minor concerns:
1. In abstract: "its expression is reduced or absent in several tumor tissues," Several types of tumor tissues?
2. Scale bar is required for all histology pictures.
3. Data from supplementary fig. 1 e, f , h and I are only collected from 3 mice.
Author Response
The authors addressed all my concerns. However, I think it is better if the authors can compare the response of KO and WT BM cells to inflammatory cytokine treatment in CFU assays.
We would like first to express our gratitude for the reviewer for his continuous interest in our manuscript and all his inputs, and although the suggested experiment would generate interesting data, however, we plan to address the role of infection and the immune response in subsequent publications.
Minor concerns:
- In abstract: "its expression is reduced or absent in several tumor tissues," Several types of tumor tissues?
We thank the review for his suggestion, and we have added the word ‘’types of’’ to the abstract of the manuscript.
- Scale bar is required for all histology pictures.
We agree with the reviewer, and we added a scale bar to the histology images in Fig1b and note the size of the scale bar in the figure legend.
- Data from supplementary fig. 1 e, f , h and I are only collected from 3 mice.
We have amended the Sup. Figure/Figure legend accordingly and added the data from two more mice.